# Cocktail, a Computer Program for Modelling Bacteriophage Infection Kinetics

**DOI:** 10.3390/v14112483

**Published:** 2022-11-09

**Authors:** Anders S. Nilsson

**Affiliations:** Department of Molecular Biosciences, The Wenner-Gren Institute, Stockholm University, SE-106 91 Stockholm, Sweden; anders.s.nilsson@su.se

**Keywords:** bacteriophage, infection kinetics, mathematical modelling, computer program, phage therapy

## Abstract

Cocktail is an easy-to-use computer program for mathematical modelling of bacteriophage (phage) infection kinetics in a chemostat. The infection of bacteria by phages results in complicated dynamic processes as both have the ability to multiply and change during the course of an infection. There is a need for a simple way to visualise these processes, not least due to the increased interest in phage therapy. Cocktail is completely self-contained and runs on a Windows 64-bit operating system. By changing the publicly available source code, the program can be developed in the directions that users see fit. Cocktail’s models consist of coupled differential equations that describe the infection of a bacterium in a vessel by one or two (interfering) phages. In the models, the bacterial population can be controlled by sixteen parameters, for example, through different growth rates, phage resistance, metabolically inactive cells or biofilm formation. The phages can be controlled by eight parameters each, such as different adsorption rates or latency periods. As the models in Cocktail describe the infection kinetics of phages in vitro, the program is primarily intended to generate hypotheses, but the results can however be indicative in the application of phage therapy.

## 1. Introduction

The ability of bacteriophages to kill bacteria has attracted increased interest in recent times. This is probably partly due to reports from clinics where they have been used for decades as bactericidal agents in the treatment of various infections, so-called phage therapy, but more recently also to a large number of studies of isolated phages (short for bacteriophages) which in laboratory experiments have been shown to be able to effectively kill bacteria in vitro, including antibiotic-resistant strains [1,2]. The increasing interest has also led to experimental treatments of severe infections, being carried out at other clinics than where they have traditionally been used for a long time [3,4]. The outcomes of clinical treatments are nevertheless difficult to predict, depending on many factors. Both bacteria and phages show an enormous variation between different clones and can change during the course of treatment, which together with largely unknown pharmacokinetics and pharmacodynamics results in treatments not being comparable and the effect of a certain treatment not being predictable. Although the variation between clinical experiments is very large, the combined results of them suggest that there is reason to continue studying how phage treatments against bacterial infections can be optimised.

A large number of mathematical models have been developed to study the dynamics between host organisms and their parasites, and bacteria–phages are no exception. Depending on the question, models use everything from evolutionary game theory (EGT), for example, on the evolution of the life cycle of phages, or networks of bacteria’s genetic changes after a bacteriophage infection (flux-balance analysis, FBA), but the most common bacteria–bacteriophage models are reaction kinetic models [5] (and references therein). These models essentially consist of coupled differential equations that describe the changes in the bacterial and bacteriophage population sizes over time in a vessel given a set of parameters. In their simplest form, the models describe the titre of susceptible, infected and resistant bacteria (classic SIR models) as well as the titre of the infecting bacteriophage. As mentioned above, bacteria and phages are self-replicating entities that can both change genetically, in the case of bacteria also phenotypically, through altered transcription [6,7]. Although simple models can provide valuable information, many models are more elaborate (for a review, see [8]). Several authors have contributed to the development of models that describe the dynamics of systems with, for example, more than one type of virulent bacteriophage, the formation of protective biofilms by bacteria or degradation of phages [9,10,11,12,13]. If models are also to be able to describe the dynamics in vitro, for example, during phage therapy of a human bacterial infection, it is also necessary to take into account the additional complexities that arise in interaction with human tissues and cells, for example, synergies between the immune system and phages, in the killing of bacteria or neutralisation of phages and phagocytosis by macrophages [14,15,16,17]. However, many of these in vitro interactions are still largely unknown in detail [18].

The purpose of the Cocktail program is to model the infection dynamics of one, or a combination of two, phage(s) infecting a bacterial species under varying relevant parameter settings, and in the latter case, to some extent, the interference between two phages infecting at the same time. The aim is to supply an easier way to carry out modelling in phage infection biology, e.g., for a better understanding of the complex dynamics during phage therapy. Although the program is easy to use, the most important parameters are included. The mathematical models are based on the basic models described, for example, by Levin et al. [10], Gill [13], Lenski [19], or Levin and Bull [20] and some of the additions discussed by Abedon [11]. It is thus possible in the program to also study the effect of metabolically inactive cells or biofilm formation as well as the decay of bacteria and phages, analogous to the action of an immune response (Figure 1).

As with all models, there has to be a balance between reality and generality. Hence, it is important to stress that some parameters, e.g., temperature, pH, release of nutrients from lysed bacteria or phages binding to cell debris, are not included in the program and others, e.g., modelling of the dynamics of an immune defence or cells in biofilm, are simplified. Therefore, the program should be seen as a tool for inducing hypotheses about the population dynamics of bacteria and phages during a phage infection and not exact predictions. This is especially important to stress regarding phage therapy experiments. The program does not consider the in vitro pharmacokinetics and pharmacodynamics of phages. Hypotheses will always need to be tested experimentally. Mathematical modelling of phage infection in a chemostat may however set boundaries to what can be expected while it can reflect the dynamics under ideal conditions (e.g., constant nutrient supply and agitation).

The following information describes the models and calculations in more detail. The basic condition is a bacterial population, growing in a vessel in a constant volume of nutrient, which can become infected by phages at varying titres and times. Although the volume is constant in such a chemostat, there could be an inflow and outflow of nutrients, and an outflow of bacteria and phages.

## 2. Materials and Methods

An overview of symbols used for all bacteria, phages and parameters can be found in Table 1. In the program, values can in general be given with three significant digits. If the input should be an integer, it can be given either as that or in scientific notation, e.g., 1,000,000 or 1.0 × 10^6^. Real numbers should be given either in decimal or scientific format with a point as the decimal separator. If the wrong format is used, values are auto corrected in most cases. Hovering with the mouse pointer above a box displays a hint on which values can be given. The tab key can be used for jumping between boxes and it is also possible to use the up and down arrows in some boxes to increase or decrease values in fixed steps. Please note that some parameter values should be given per hour and others per minute, as shown by the default values (Table 1). Additionally, note that many parameters are represented with their average values despite in reality being distributed in time or size, e.g., the latent period will vary from cell to cell and not all cells will generate the same burst size (produce exactly the same number of phages).

### 2.1. Bacteria

For most bacteria, the rate of growth mainly depends on the concentration of nutrients (physical factors, e.g., temperature and the presence of various gases are of course also important). In a closed system, while nutrients are consumed by the bacteria, their concentration decreases and growth slows. Seen over time, in such cases the bacterial growth becomes a logistic function of the concentration of nutrients. One of the most widely used relationships between growth rate and nutrient concentration was formulated by Monod [21]; the growth of bacteria depends on three parameters, µ = µ_max_ × s/(K_s_ + s) where the growth rate, µ, is a function of the maximum growth rate, µ_max_, a constant, K, and the concentration of nutrients, s. This is also the basic growth function in the Cocktail program where most parameter symbols and default parameter settings are taken from Lenski [19] (Table 1). In the program, the max growth rate per hour is denoted as ψ, a real number between 0 and 1.5 and the Monod half-saturation constant, K in µg/mL, is a real number between 0.01 and 100. The reason for allowing this large span is the observed variation among strains and experimental conditions when assessing the half-saturation constant, but the values are usually between 0.1 and 10 µg/mL [22]. The concentration of nutrients varies in a chemostat while the nutrients are consumed by bacteria and where there is an inflow and outflow. It is denoted C in the program and the concentration of nutrients over time equation is described in more detail below. While mutated bacteria may suffer from reduced fitness, the growth rate of bacteria that have become resistant to either one or both phages can be entered separately. Bacteria can become resistant to phage infection by mutation but only at cell division when new cells are produced. The mutation rates should also be written as decimal numbers or in scientific notation. Note that bacteria becoming resistant to both phages, A and B, occurs at a rate being the product of the rate of mutation to resistance against A and B, respectively, and does not need to be specified. It is also possible that the starting population of bacteria may contain resistant bacteria to either one or both phages. These frequencies are also entered either as decimal numbers or in scientific notation as above.

Bacteria can also decay from natural causes, and not just die from a phage infection. In an in vitro situation, this would be, e.g., from neutralisation by antibodies or phagocytosis. This results in an exponential decay N_t_ = N_0_ × e^−γt^ where N_0_ is the number of bacteria at time 0 and N_t_ at time t and γ is the decay rate constant. The decay is calculated as N_t+1_ = N_t_ − (N_t_ × γ) in the equations below while this equals N_0_ × e^−γt^. γ is quite small for bacteria growing in a chemostat, half of the population has decayed at t = ln(2)/γ, and values of γ is generally around 0.02 per hour in nature [23,24]. The decay rate per hour, γ, should be given as a real number between 0 and 1. Note that new bacteria resulting from cell division each generation is excluded from decay.

### 2.2. Resources

The addition of nutrition is necessary for the growth of bacteria. The concentration of nutrients is regulated by the flow into and out of the system at a rate of ω turnovers/hour. The starting concentration, C_0_ in µg/mL, can be set independently from the concentration of nutrients continuously flowing into the system, C in µg/mL. The flow in and out of the system causes dynamic changes in the system while it is not just nutrients flowing out but also bacteria and phages. The conversion efficiency, the resources used by one dividing cell, is denoted as ε and given in µg/cell as a decimal number or in scientific notation, e.g., 1.0 × 10^−6^. The equation for dC/dt is shown below in the following text.

### 2.3. Phages

Two virulent phages, A and B, can infect the bacterial population in different titres, adsorption rates, and at different times while it is possible to let the bacterial population grow for some time and add phages at three different time points. As with the bacteria, phages can decay and be washed out of the system. Phages inactivated by binding to lysed cell debris cannot be entered separately but must be included in the decay. Phage populations grow by two additional parameters. When the latent period comes to an end, infected bacteria burst open producing the burst size number of phages. Note that a fast phage of one type, e.g., A, infecting a bacterium already infected with a slower phage, e.g., B, will produce only type A phages if its latent period is shorter than the remaining time of phage B’s latent period. If only one phage is chosen to infect, the added titre of the other phage should be set to 0. Additionally, if one wants to study a nonproductive infection, the burst size should be set to zero.

### 2.4. Model Settings

A phage infection starts with the adsorption of the phage to receptors on the bacterial surface. The adsorption can be set to happen by different mathematical models described in the following text. In short, phages can adsorb one by one per unit time as described in the primary adsorption “Standard” model below or several phages at once per unit time according to the Poisson probability of infection. The “Poisson” setting hence allows for multiple adsorption proportional to the multiplicity of infection. It is also possible to adjust both models to allow adsorption to uninfected and non-resistant cells only or to all susceptible and non-resistant cells irrespective if they have already been infected. These secondary adsorption alternatives are chosen with the settings “Uninfected” or “Susceptible”. An overview over which types of cells that get adsorbed and infected by which phages is given in Table 2. Details and the mathematical background to all models is given in the text that follows.

#### 2.4.1. Primary Adsorption: Standard Model

Bacteria can divide and the population grow, but bacteria can also decay, be completely resistant or mutate to resistance against phage infection (Section 2.4.3), hide in a refuge (Section 2.4.4) as well as become infected by phages and lyse from the infection. In the Cocktail program, this results in the possibility of thirteen types of bacteria being present in the system at the same time (Table 1). As mentioned, phages A and B can infect at different times and by different adsorption rates, and after varying latent periods lyse infected bacteria resulting in a burst of phages of different size. Bacteria and phages can both flow out of the system. Starting with the concentration of nutrients, this results in the following basic equations:(1)$$\frac{dC}{dt}=\left(C_{0}-C\right)\omega -\frac{\varepsilon N\psi C}{K+C}$$

The first Equation (1) describes the nutrient content in the system over time where *C* is the concentration of nutrient in µg/mL (*C*_0_ is the start concentration), ω is the flow rate in turnovers/h, ε is the resource consumption by one bacterium in µg/cell and *ψ* is the specific growth rate of bacteria per hour. N stands for the S and R types of bacteria (*S*, *R_A_*, *R_B_*, *R_AB_*) as the infected bacteria are supposed to cease both to consume nutrients and to divide. K is the Monod half-saturation constant in µg/mL (the concentration of nutrients resulting in half the maximum growth of the bacteria). For simplicity, ε and *K* are the same for all dividing bacteria, but *ψ* can vary and be different for phage-resistant bacteria.
(2)$$\frac{dS}{dt}=\frac{S\psi_{S}C}{K+C}-\frac{(\mu_{A}+\mu_{B}+\mu_{AB)}S\psi_{S}C}{K+C}+\rho S_{r}-\sigma S-\gamma S-\delta_{A}SA-\delta_{B}SB-\omega S$$

This second Equation (2) describes the growth and losses of susceptible bacteria. The losses over time are due to bacteria mutating to become resistant against phage *A* or *B* or both at a specific rate of *μ* per cell division (*µ_AB_* is calculated as *µ_A_* × *µ_B_*). This means that it is only divided bacteria that mutate. Bacteria can also move into a refuge population, *S_r_*, at a rate of *σ* where they may or may not be adsorbed by phages, and move out of the refuge at a rate of *ρ*. Bacteria can also decay or be neutralised at a rate γ, becoming infected by phages at the adsorption rate *δ*, and finally be washed out of the system at a rate of *ω*. The equations for phage-resistant bacteria become:(3)$$\frac{dR_{A}}{dt}=\frac{R_{A}\psi_{R_{A}}C}{K+C}+\mu_{A}S+\rho R_{rA}-\sigma R_{A}-\gamma R_{A}-\mu_{B}R_{A}-\delta_{B}R_{A}B-\omega R_{A}$$
(4)$$\frac{dR_{B}}{dt}=\frac{R_{B}\psi_{R_{B}}C}{K+C}+\mu_{B}S+\rho R_{rB}-\sigma R_{B}-\gamma R_{B}-\mu_{A}R_{B}-\delta_{A}R_{B}A-\omega R_{B}$$
(5)$$\frac{dR_{AB}}{dt}=\frac{R_{AB}\psi_{R_{AB}}C}{K+C}+\mu_{AB}S+\rho R_{rAB}-\sigma R_{AB}-\gamma R_{AB}+\mu_{A}R_{B}+\mu_{B}R_{A}-\omega R_{AB}$$

Phage resistance is intended to result in complete blocking of adsorption by the phage. Hence, a bacterium resistant to phage *A* can mutate to become resistant to phage B as well but also be infected by phage *B*, and vice versa. Needless to say, a bacterium resistant to both phages (Equation (5)) cannot become infected at all.

Infected bacteria (Equations (6)–(8)) are thought not to consume resources or divide and not to be part of the refuge population of cells. Cells in the refuge (see Equations (26) and (27) below) is simulating the presence of either metabolically inactive cells or cells in biofilm and inhibited phage propagation. Phages can however adsorb to different classes of bacteria depending on the secondary adsorption mode setting. In the “Uninfected” setting, phages are adsorbing to uninfected bacteria only, which is common in basic mathematical models, where phage A adsorbs only to *S*, *I_B_*, and *R_B_*. In the “Susceptible” setting, on the other hand, phages are allowed to adsorb to already-infected bacteria as well, i.e., A adsorbs to *S*, *I_A_*, *I_B_*, *I_AB_*, *R_B_* and *R_BIA_* (referred to as secondary adsorption [11]). In addition to this, *A* can also adsorb to the *S_r_* and *R_rB_* cells in the refuge if the “Standard” primary adsorption model is applied, the “Planktonic” mode is set and the rate of cells in and out of the refuge are given values. Phage B adsorbs to *S_r_* and *R_rA_* as well with this setting. Cells in the refuge will however not be infected, only adsorbed by phages.
(6)$$\frac{dI_{A}}{dt}=\delta_{A}SA-\gamma I_{A}-\delta_{B}I_{A}B-I_{A\left(t-l_{A}\right)}-\omega I_{A}$$
(7)$$\frac{dI_{B}}{dt}=\delta_{B}SB-\gamma I_{B}-\delta_{A}I_{B}A-I_{B\left(t-l_{B}\right)}-\omega I_{B}$$
(8)$$\frac{dI_{AB}}{dt}=\delta_{A}I_{B}A+\delta_{B}I_{A}B-\gamma I_{AB}-I_{AB\left(t-l_{B}\right)}-I_{AB\left(t-l_{A}\right)}-\omega I_{AB}$$

The expressions $I_{A\left(t-l_{A}\right)}$, $I_{B\left(t-l_{B}\right)}$, $I_{AB\left(t-l_{B}\right)}$ and $I_{AB\left(t-l_{A}\right)}$, part of the delayed differential Equations (6)–(8), are all describing the loss of infected bacteria due to lysis, after the latency time *l_A_* or *l_B_*. A bacterium simultaneously infected by two phages will lyse at time *l_A_* if the latency of phage *A* is shorter than the latency time for phage *B*, *l_B_*. This results in that $I_{AB\left(t-l_{B}\right)}$ becomes zero for a bacterium when $I_{AB\left(t-l_{A}\right)}$ becomes positive. One of $I_{AB\left(t-l_{A}\right)}$ and $I_{AB\left(t-l_{B}\right)}$ subsequently always becomes zero. A bacterium infected with phage *B* at time t_B_ and superinfected with phage *A* at time t_A_ will lyse and produce phages of type *A* only if *t_A_* + *l_A_* is shorter than the remaining time to lysis caused by phage *B*, i.e., *t_A_ + l_A_ < t_A_ + l_A_ − t_B_ + l_B_ = t_A_ < t_A_ − t_B_ + l_B_*. This means that in an infection with both phages, *A* and *B* will interfere with each other and not produce the number of phages expected from single and separate infections by *A* or *B*. Other interference between phages, e.g., by actively impeding the other phage’s transcription or replication, is not considered.

Finally, bacteria that are resistant to infections by one phage can become infected with the other phage (Equations (9) and (10)).
(9)$$\frac{dR_{BIA}}{dt}=\delta_{A}R_{B}A-\gamma R_{BIA}-\delta_{A}R_{BIA\left(t-l_{A}\right)}-\omega R_{BIA}$$
(10)$$\frac{dR_{AIB}}{dt}=\delta_{B}R_{A}B-\gamma R_{AIB}-\delta_{B}R_{AIB\left(t-l_{B}\right)}-\omega R_{AIB}$$

Titres of phages *A* and *B* can grow through the release of phages from all types of infected bacteria (13 and 14). After the phage specific latency periods mentioned above, each bacterial cell gives rise to the number of phages equal to the phage’s burst size, β. Phages will be lost by adsorption to the bacteria mentioned above, and adsorbed phages representing phages bound to bacteria, is denoted *P_A_* and *P_B_*, respectively (11 and 12). Phages can also decompose at a rate of φ, and be washed out at a rate of ω. With the no secondary adsorption “Uninfected” setting:(11)$$P_{A}=\delta_{A}A\left(S+I_{B}+R_{B}\right)\;\mathrm{and}\;P_{B}=\delta_{B}B\left(S+I_{A}+R_{A}\right)$$

Additionally, with the “Susceptible” setting:(12)$$P_{A}=\delta_{A}A\left(S+I_{B}+R_{B}+I_{A}+I_{AB}+R_{BIA}\right)\;\mathrm{and}\;P_{B}=\delta_{B}B\left(S+I_{A}+R_{A}+I_{B}+I_{AB}+R_{AIB}\right)$$

The change in phage titres will hence be:(13)$$\frac{dA}{dt}=\beta_{A}\left(I_{A\left(t-l_{A}\right)}+I_{AB\left(t-l_{A}\right)}+R_{BIA\left(t-l_{A}\right)}\right)-P_{A}-\varphi_{A}A-\omega A$$
(14)$$\frac{dB}{dt}=\beta_{B}\left(I_{B\left(t-l_{B}\right)}+I_{AB\left(t-l_{B}\right)}+R_{AIB\left(t-l_{B}\right)}\right)-P_{B}-\varphi_{B}B-\omega B$$

#### 2.4.2. Secondary Adsorption: Poisson

Only uninfected bacteria can be infected by phages in the “Standard” model with the “Secondary adsorption Uninfected” setting ($\delta_{A}SA$ in the equations). This is in many cases a good approximation but neglects that several phages may adsorb to a single bacterial cell, assuming that the number of cell receptors is not limited, i.e., multiple adsorption [11]. Phages adsorbed to cells will follow Poisson probabilities. While more phages per bacterium can infect, this is referred to as MOI_actual_ in contrast to MOI_input_ [11,25]. Bound phages will hence be:(15)$$P_{t+1}=\left(1-e^{-\delta M_{t}}\right)P_{t}$$

Here, *P* is the titre of phages and *M* is the sum of all bacteria that can be adsorbed by a particular phage. These are denoted *M_A_* for phage *A* and are bacteria *S*, *I_B_*, and *R_B_*, in the “Uninfected” adsorption mode and S, I_A_, I_B_, I_AB_, R_B_ and R_BIA_ with the “Susceptible” setting, i.e., the same sets of bacteria as in the “Standard” primary adsorption model. M_B_ is accordingly bacteria *S*, *I_A_*, and *R_A_*, and *S*, *I_B_*, *I_A_*, *I_AB_*, *R_A_* and *R_AIB_*, respectively. While more phages than one may adsorb to a cell, infected bacteria will equal:(16)$$I_{t+1}=(1-e^{-\frac{P_{t}}{M_{t}}})M_{t}$$

Taking Equation (16) into account, the equations describing the change in uninfected bacterial titres (17–19) will have to change to:(17)$$\frac{dS}{dt}=\frac{S\psi_{S}C}{K+C}-\frac{(\mu_{A}+\mu_{B}+\mu_{AB)}S\psi_{S}C}{K+C}+\rho S_{r}-\sigma S-\gamma S-((1-e^{-\frac{P_{A}}{M_{A}}})M_{A})-((1-e^{-\frac{P_{B}}{M_{B}}})M_{B})-\omega S$$

The equations for resistant bacteria will change accordingly:(18)$$\frac{dR_{A}}{dt}=\frac{R_{A}\psi_{R_{A}}C}{K+C}+\mu_{A}S+\rho R_{rA}-\sigma R_{A}-\gamma R_{A}-\mu_{B}R_{A}-((1-e^{-\frac{P_{B}}{R_{A}}})R_{A})-\omega R_{A}$$
(19)$$\frac{dR_{B}}{dt}=\frac{R_{B}\psi_{R_{B}}C}{K+C}+\mu_{B}S+\rho R_{rB}-\sigma R_{B}-\gamma R_{B}-\mu_{A}R_{B}-((1-e^{-\frac{P_{A}}{R_{B}}})R_{B})-\omega R_{B}$$
(20)$$\frac{dR_{AB}}{dt}=\frac{R_{AB}\psi_{R_{AB}}C}{K+C}+\mu_{AB}S+\rho R_{rAB}-\sigma R_{AB}-\gamma R_{AB}+\mu_{A}R_{B}+\mu_{B}R_{A}-\omega R_{AB}$$

If the infection is by the primary adsorption option “Poisson” and bacteria are chosen to be in the planktonic refuge and entered at a certain rate (see Equation (26) below), phages are additionally also adsorbing to the bacteria in the refuge. Phage *A* will additionally adsorb to *S_r_* and *R_rB_* cells and phage *B* to *S_r_* and *R_rA_* cells. However, this is not the case if the last-in-first-out (“LIFO”) type of refuge cells is selected (27). Bound phages, *A* and *B*, are again denoted *P_A_* and *P_B_*, respectively. In the Poisson mode, this results in Equations (21)–(25) for infected bacteria:(21)$$\frac{dI_{A}}{dt}=((1-e^{-\frac{P_{A}}{M_{A}}})M_{A})-\gamma I_{A}-\mu_{B}I_{A}-((1-e^{-\frac{P_{A}}{I_{B}}})I_{B})-I_{A\left(t-l_{A}\right)}-\omega I_{A}$$
(22)$$\frac{dI_{B}}{dt}=((1-e^{-\frac{P_{B}}{M_{B}}})M_{B})-\gamma I_{B}-\mu_{A}I_{B}-((1-e^{-\frac{P_{B}}{I_{A}}})I_{A})-I_{B\left(t-l_{B}\right)}-\omega I_{B}$$
(23)$$\frac{dI_{AB}}{dt}=((1-e^{-\frac{P_{A}}{I_{B}}})I_{B})+((1-e^{-\frac{P_{B}}{I_{A}}})I_{A})-\gamma I_{AB}-I_{AB\left(t-l_{B}\right)}-I_{AB\left(t-l_{A}\right)}-\omega I_{AB}$$
(24)$$\frac{dR_{BIA}}{dt}=((1-e^{-\frac{P_{A}}{R_{B}}})R_{B})-\gamma R_{BIA}-R_{BIA\left(t-l_{A}\right)}-\omega R_{BIA}$$
(25)$$\frac{dR_{AIB}}{dt}=((1-e^{-\frac{P_{B}}{R_{A}}})R_{A})-\gamma R_{AIB}-R_{AIB\left(t-l_{B}\right)}-\omega R_{AIB}$$

The expression of phages *A* and *B* lost by adsorption to bacteria is the same as in the “Standard” model, described in Equations (13) and (14), but bound phages *P_A_* and *P_B_* is calculated differently as shown in Equation (15) above.

It should be pointed out that the difference between the “Standard” mode of infection and the “Poisson” mode is obviously small at a MOI around 1. It is only when there are phages in excess, a probability of more than one phage infecting a bacterium, that a difference may be observed as a more rapid loss of adsorbed phages.

#### 2.4.3. Resistance Mutation

Dividing bacteria can mutate at a rate of µ, and the mutant frequencies in the population can be calculated in two different ways. In the “Deterministic” mode, each class of newly divided bacteria contains N × µ mutants, where N is the number of newly divided bacteria. With the “Stochastic” alternative, mutations are introduced by random sampling from a Poisson distribution having a mean of N × µ = λ if λ ≤ 10 or from sampling a normal distribution if λ > 10. The normal distribution, (λ;√ λ), is generated by the Box-Müller algorithm and all random numbers are generated by a Mersenne Twister algorithm. This results in good, but somewhat slow, generation of pseudo random numbers but this does not have a great impact on the overall program performance. When stochastic mutation is active, results will of course vary from run to run. With small numbers of divided bacteria per millilitre, for example, 10^5^ bacteria, and a mutation rate of 10^−7^, the resulting frequency of mutants is bound to be very low. There would be only 0.01 mutant bacteria in the population and these would be eliminated if the option of rounding off values below one is activated. In the program, resistance is modelled as affecting the adsorption and regarded as complete. Therefore, resistant bacteria do not adsorb any phages.

#### 2.4.4. Refuge Cells

The refuge population is simulating the existence of metabolically inactive cells, with the “Planktonic” setting, or cells forming biofilm with the last-in-first-out (“LIFO”) setting. The “LIFO” setting means that the last cells that entered the refuge are reintroduced to the normal pool of cells followed by the next to last cells and so forth. The rate of cells moving into the refuge can be set to at most 0.01, which means that 1% of the current population will enter the refuge per minute. Another limitation is that there has to be more than 10 cells outside of the refuge. In such a case, only 0.1 cells enter the refuge. If only whole cells should be allowed to enter, the round off <1 option should be chosen. Both boxes, the rate in/min and rate out/min must be given a value in order to activate the refuge cells models. While in the refuge, these cells are not dividing and cannot produce phages or mutate and become resistant which they can become upon reintroduction to the normal pool of cells. Infected bacteria (*I_A_*, *I_B_*, *I_AB_*, *R_AIB_*, *R_BIA_*) are not part of either refuge population, as these cells will lyse in any event. In the “Planktonic” mode, cells can be flushed out of the system or decay whereas in “LIFO” mode cells are thought to be metabolically inactive and sessile until reintroduced. Planktonic refuge cells:(26)$$\frac{dS_{r}}{dt}=\sigma S-\rho S_{r}-\gamma S_{r}-\omega S_{r}\;;\;\frac{dR_{rA}}{dt}=\sigma R_{A}-\rho R_{rA}-\gamma R_{rA}-\omega R_{rA}\;\frac{dR_{rB}}{dt}=\sigma R_{B}-\rho R_{rB}-\gamma R_{rB}-\omega R_{rB}\;;\;\frac{dR_{rAB}}{dt}=\sigma R_{AB}-\rho R_{rAB}-\gamma R_{rAB}-\omega R_{rAB}$$

Cells in the last-in-first-out (LIFO) mode:(27)$$\frac{dS_{r}}{dt}=\sigma S-\rho S_{r}\;;\;\frac{dR_{rA}}{dt}=\sigma R_{A}-\rho R_{rA}\;\frac{dR_{rB}}{dt}=\sigma R_{B}-\rho R_{rB}\;;\;\frac{dR_{rAB}}{dt}=\sigma R_{AB}-\rho R_{rAB}$$

#### 2.4.5. Time Step Size

Calculations of differential equations in Cocktail are done using Euler’s method, taking the input values as the initial values for calculating the values numerically after the chosen time interval, set either as one minute or as a 30-, 15- or 5-second time step size. At large time intervals, other methods for solving differential equations result in smaller errors, but the differences between methods become smaller and smaller as the step size (time interval) decreases. Hence, running the program with a step size of one minute results in a larger discretisation error than with a five-second step size, but is considerably faster. On the other hand, the program speed depends mainly on the length of the phages’ latent periods and the rates of bacteria into and out of refuge populations. These are stored on arrays in dynamic memory that need to be recalculated in each step, which will slow down the program performance if a long running time is set. However, there is virtually no time difference between short and long time step sizes if a short running time and no refuge cells are chosen. If accuracy and a small discretisation error is preferred, step size should be set to five-second intervals.

## 3. Results

Examples of Cocktail outputs can be studied by running the example data files provided as Appendix A. The file Bohannan_Lenski_1997_Fig 3B.ctl contains input parameter values and a comparison to a chemostat experiment where the authors found that the titre of Escherichia coli bacteria and an infecting T4 phage can oscillate over time [26]. The phage titre decreases over time when there are very few bacteria to infect which in turn results in a higher titre of bacteria and so forth (Figure 2A). Coexistence can theoretically be shown to occur at higher bacterial titres as well. Running the parameter settings in the file Lenski_1988_Fig_2a.ctl from Lenski [19] results in oscillations leading to stable coexistence of bacteria in a titre of about 10^7^ and phage titres being around 10^9^ (Figure 2B). However, a fourfold increase in the concentration of nutrients, from 25 to 100 µg/mL results in increasingly large oscillations, but as in the first case, bacteria never become extinct (Figure 2C).

It is also possible to analyse more complex problems and formulate hypotheses that later can be evaluated experimentally. The last example describes two phages in a cocktail with different latent periods, added in the same titres, and at the same time to the bacteria. Most parameters were entered with their default values (Table 1). Parameters set to different values were the bacteria’s starting titre, 1 × 10^8^, and the program was executed with the standard model where phages only adsorb to uninfected cells, mutations are set to be deterministic and no cells were allowed to be resistant to either phage A or B or both, metabolically inactive or form biofilm. The log_10_ option was selected for the output. While log_10_(0) = −∞, a titre of 0 is represented as −16 (log_10_ of 10^−16^). The data can be retrieved by running the file Fig_2D.ctl. The result showed that bacteria resistant to phage A will disappear from the system within an hour (Figure 2D). Phage B has a shorter latent period, and has outcompeted phage A, by infecting most of the susceptible bacteria. The mutation rate of bacteria becoming resistant to both phages was set to 10^−7^ × 10^−7^ (the product of mutation rate for resistance to A and B, respectively) which resulted in a low titre of double resistant cells, as the bacteria had a good supply of nutrients and were able to divide, but the titre of such cells will grow to about a thousand cells/mL in 48 h. Allowing resistant cells from the beginning, in the start population, results in higher titres of such cells.

## 4. Technical Information

Cocktail runs on Windows 64-bit systems. The program interface (Figure 1) is in English, but some instructions may turn up in the language set on your computer when Windows DLLs are called (e.g., file dialogs). There is no support for other languages in Cocktail. The program is developed in Object Pascal from the Free Pascal Team (Free Pascal: A 32-, 64- and 16-bit professional Pascal compiler. Version 3.2.0. URL https://www.freepascal.org. RRID:SCR_014360), accessed on 28 September 2022, using the Lazarus IDE and libraries developed by the Lazarus Team (Lazarus: The professional Free Pascal RAD IDE. Version 2.0.10. URL http://www.lazarus-ide.org. RRID:SCR_014362, accessed on 28 September 2022). The IDE, compiler and program libraries can be downloaded from: https://www.lazarus-ide.org/index.php?page=downloads, accessed on 28 September 2022. Source code files (cocktailunit1.pas, cocktailunit2.pas, cocktailunit1.lfm, cocktailunit2.lfm, Cocktail.lps, Cocktail1.lpr, Cocktail1.res, Cocktail.dbg, globalvariables.pas, Cocktail.ico and the Lazarus project information file Cocktail.lpi) as well as updates will be available from GitHub at https://github.com/ASNilsson/Cocktail-phage-infection-kinetics, accessed on 28 September 2022.

The Cocktail program and source code files are distributed under the license Creative Commons Attribution-NonCommercial-ShareAlike 4.0 International License. In short, this means that it is free for everyone to use, to modify the source code, build upon the program or code, and free to distribute in any medium. However, you must give appropriate credit and a link to the license. If changes were made to the program or code, these must be specified, and distribution of modifications must be under the same license. It is not allowed for anyone to use any part of the program or code for commercial purposes. A short description of the license can be found at: https://creativecommons.org/licenses/by-nc-sa/4.0/, accessed on 28 September 2022. The license and program version number can be found by double clicking anywhere in the Cocktail parameter settings window.

The results of the program can be saved as charts, in PNG or SVG graphics file formats, and/or as a .ctl data file. The items in the .ctl file constitute the complete settings for running the program. The file format is a plain .txt file, but note that the format is fixed as in the example .ctl file. Moving items to another position in the file will inevitably result in a file error. A comma (,) is often used as a delimiter when more than one item is to be found on a line. The advantage of this is that the file can contain a short label of each of the items which makes reading and editing a file much easier. The disadvantage is that omitting a comma, or using another delimiter, will result in a file error. Selected output parameters should however be surrounded by at least one blank in the list following the label “Output parameters:”, e.g., 1 11 14 (note the blank after the last number). A .ctl file can easily be created by running the program and saving the result by clicking on the “Save” button at the bottom of the result window. Editing such a .ctl file that has been shown to work as a template, and saving it under a new name, is a good idea. Double clicking on a .ctl file will open a new instance of the program provided that a link to the program has been established in the Windows “How do you want to open this file?” dialog by marking the Cocktail program and checking the “Always use this app to open .ctl files” box.

## Figures and Tables

**Figure 1 viruses-14-02483-f001:**
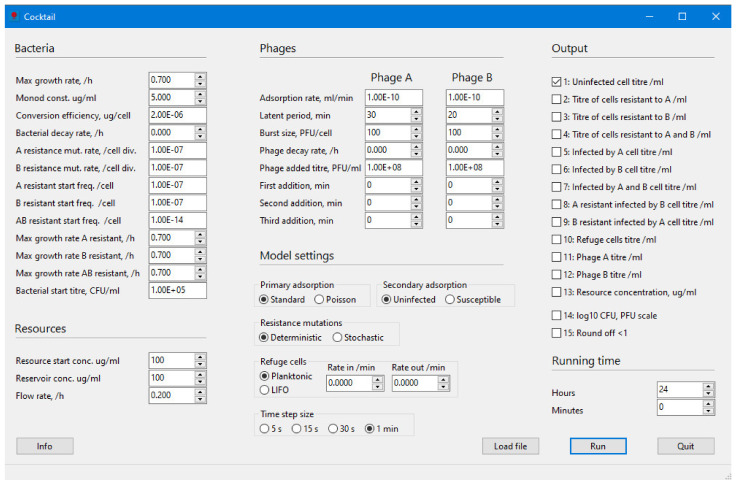
The Cocktail user interface.

**Figure 2 viruses-14-02483-f002:**
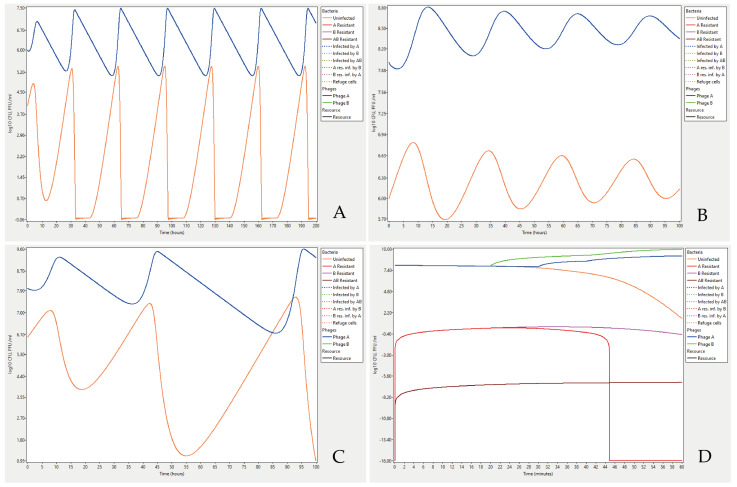
Example output graphs from the Cocktail program: (**A**) *Escherichia coli* bacteria infected with phage T4 in a chemostat where both the bacteria and the phage titre fluctuate under certain conditions. The parameter settings were as in [26] with the exception of the resource density being set to 1.0 instead of 0.5 mg/L (µg/mL) in the chemostat, and the time step size set to 1 min instead of 3 min. The run was for the first 200 h of the original chemostat experiment (before development of bacterial phage resistance). Complete parameter settings can be found in the file Bohannan_Lenski_1997_Fig 3B.ctl in the Appendix A. (**B**) Oscillations of bacterial and phage can theoretically exist even at higher titres as shown by Lenski [19]. Bacteria in a titre of 10^6^ cell forming units is infected with virulent phages with a burst size of 100 and in a titre of 10^8^. The stability of the system depends on a low concentration of nutrients, 25 µg/mL. The parameter settings can be found in the Appendix A file Lenski_1988_Fig_2a.ctl. (**C**) As the concentration of nutrients increases four times, the titres of both bacteria and phages shift. This results in increasingly large oscillations where bacterial titres are reduced to a few cells every cycle, but they never become extinct. The settings are from the file Lenski_1988_Fig_2b.ctl. in the Appendix A. (**D**) An example of bacteria at high titres simultaneously infected by two phages with different infection characteristics. Bacteria that mutate and become resistant to either one of the phages are eventually lost and non-resistant bacteria slowly become extinct but replaced by bacteria resistant to both phages. See text for more details. Settings from the file Fig_2D.ctl in the Appendix A.

**Table 1 viruses-14-02483-t001:** Symbols and parameters.

		Start Values		
Symbol	Description	Default	Allowed Range	Unit
Bacteria				
*S*	Uninfected, susceptible bacteria	1 × 10^5^	10^−1^ × 10^12^	CFU/mL
*I_A_*	Bacteria infected by phage *A*	-	-	|
*I_B_*	Bacteria infected by phage *B*	-	-	|
*I_AB_*	Bacteria infected by phages *A* and *B*	-	-	|
*R_A_*	Bacteria resistant to phage *A*	1 × 10^−7^	0–1 × 10^−2^	|
*R_B_*	Bacteria resistant to phage *B*	1 × 10^−7^	0–1 × 10^−2^	|
*R_AB_*	Bacteria resistant to phages *A* and *B*	1 × 10^−14^	0–1 × 10^−6^	|
*R_AIB_*	Bacteria resistant to *A* infected with *B*	-	-	|
*R_BIA_*	Bacteria resistant to *B* infected with *A*	-	-	|
*S_r_*	Susceptible bacteria in a refuge	0	-	|
*R_rA_*	Bacteria resistant to *A* in a refuge	-	-	|
*R_rB_*	Bacteria resistant to *B* in a refuge	-	-	|
*R_rAB_*	Bacteria resistant to *AB* in a refuge	-	-	CFU/mL
Parameters				
*ψ*	Growth rate of *S*	0.7	0–1.5	/h
*K*	Monod constant	5.0	0.01–100	µg/mL *
*ε*	Resource for division of one bacterium	2 × 10^−6^	1 × 10^−8^–1 × 10^−4^	µg/cell *
*γ*	Bacterial decay rate	0	0–1	/h
*µ_A_*	Mutation rate for resistance against *A*	1 × 10^−7^	0–1 × 10^−2^	/cell div.
*µ_B_*	Mutation rate for resistance against *B*	1 × 10^−7^	0–1 × 10^−2^	/cell div.
$\psi_{R_{A}}$	Growth rate of *R_A_*	0.7	0–1.5	/h
$\psi_{R_{B}}$	Growth rate of *R_B_*	0.7	0–1.5	/h
$\psi_{R_{AB}}$	Growth rate of *R_AB_*	0.7	0–1.5	/h
σ	Rate of bacteria into refuge	0	0–0.01	/min
ρ	Rate of bacteria out from refuge	0	0–0.01	/min
*C_0_*	Available resources from start	100	0–1000	µg/mL *
*C*	Resources flowing in from a reservoir	100	0–1000	µg/mL *
*ω*	Flow rate	0.2	0–100	/h
Phages				
Parameters				
*A*	Titre of phage *A*	1 × 10^8^	0–1 × 10^13^	PFU/mL
*B*	Titre of phage *B*	1 × 10^8^	0–1 × 10^13^	PFU/mL
*δ_A_*	Adsorption rate of *A*	1 × 10^−10^	1 × 10^−14^–1 × 10^−7^	mL/min
*δ_B_*	Adsorption rate of *B*	1 × 10^−10^	1 × 10^−14^–1 × 10^−7^	mL/min
*l_A_*	Latent period of *A*	30	1–60	min
*l_B_*	Latent period of *B*	20	1–60	min
*β_A_*	Burst size of *A*	100	0–1000	PFU/cell
*β_B_*	Burst size of *B*	100	0–1000	PFU/cell
*φ_A_*	Decay rate of phage *A*	0	0–1	/h
*φ_B_*	Decay rate of phage *B*	0	0–1	/h

* The symbol for the micro prefix, “*µ*”, is denoted by “u” in the program user interface.

**Table 2 viruses-14-02483-t002:** Adsorbed *A* and *B* phages per unit time under the primary and secondary adsorption settings.

Primary Adsorption Setting	Standard		Poisson	
Secondary Adsorption Setting	Uninfected	Susceptible	Uninfected	Susceptible
	Phages adsorb one at a time to uninfected non-resistant cells	Phages adsorb one at a time to non-resistant cells	A number of phages adsorb according to a Poisson probability with lambda = MOI to uninfected non-resistant cells	A number of phages adsorb according to a Poisson probability with lambda = MOI to non-resistant cells
Bacteria	Conceivably adsorbing phages			
*S* = Susceptible	*A* or *B*	*A* or *B*	*A* and *B*	*A* and *B*
*I_A_* = Infected by *A*	*B*	*A* or *B*	*B*	*A* and *B*
*I_B_* = Infected by *B*	*A*	*A* or *B*	*A*	*A* and *B*
*I_AB_* = Infected by *A* and *B*	-	*A* or *B*	-	*A* and *B*
*R_A_* = Resistant to infections by *A*	*B*	*B*	*B*	*B*
*R_B_* = Resistant to infections by *B*	*A*	*A*	*A*	*A*
*R_AB_* = Resistant to infections by *A* and *B*	-	-	-	-
*R_AIB_* = Resistant to infections by *A* infected with *B*	-	*B*	-	*B*
*R_BIA_* = Resistant to infections by *B* infected with *A*	-	*A*	-	*A*
*S_r_* = Susceptible planktonic bacteria in a refuge	-	*A* or *B* No infection	*A* and *B* No infection	*A* and *B* No infection
*R_rA_* = Planktonic bacteria resistant to *A* in a refuge	-	*B* No infection	*B* No infection	*B* No infection
*R_rB_* = Planktonic bacteria resistant to *B* in a refuge	-	*A* No infection	*A* No infection	*A* No infection
*R_rAB_* = Planktonic bacteria resistant to *AB* in a refuge	-	-	-	-
*S_r_* = Susceptible bacteria in a LIFO refuge	-	-	-	-
*R_rA_* = Bacteria resistant to *A* in a LIFO refuge	-	-	-	-
*R_rB_* = Bacteria resistant to *B* in a LIFO refuge	-	-	-	-
*R_rAB_* = Bacteria resistant to *AB* in a LIFO refuge	-	-	-	-

## Data Availability

Source files, data files and updates can be downloaded from https://github.com/ASNilsson/Cocktail-phage-infection-kinetics (accessed on 30 September 2022).

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
