# Peer review of "Cocktail, a Computer Program for Modelling Bacteriophage Infection Kinetics"

_viruses, 2022, doi:10.3390/v14112483_

Round 1
Reviewer 1 Report
This paper describes a program “Cocktail” for simulating bacteriophage-host interactions in a controlled environment. The program is easy to use, with a simple and attractive graphical interface, and would be useful for educational purposes and potentially to generate testable hypotheses about the phage-host dynamics in a real-world environment.
The paper is well written and clearly laid out. I am not qualified to assess the mathematical model itself, but it seems well described and I will assume that the mathematical foundation is sound.
The main flaw of the paper is that there is no attempt to compare the results of the simulation to an actual experiment. How does the program replicate phage-host dynamics in an real chemostat? Even if the author does not have access to a chemostat in his lab or institution, there must be examples in the literature to which the program’s output can be compared. As the author himself points out: "Hypotheses will always need to be tested experimentally" (line 82). The paper would be strengthened considerably if the author did just that.
The results section is vague as to the source of the numbers input to the program: “The example describes two phages in a cocktail with different latent periods.” What two phages? Where do the default values come from? I would like to see a table of phages and hosts with the actual values for absorption rates, latent periods, burst sizes, growth rates, mutation rates etc. and how the program performs with these known parameters. Otherwise, there is no way to know whether the output from the program is just a pretty plot, or actually reflecting phage-host dynamics in a chemostat, let alone in a complex, real-world scenario.
Although the manuscript does state explicitly that there are limitations to the program, it could state more clearly at the outset that the program models phage-host interactions in a chemostat. (For example, add “…in a chemostat” at the end of the first sentence of the abstract.) It is probably a stretch to imagine that the modeling will be relevant to an experimental phage therapy setting, and I would tone down this angle; I do think that the program could serve as an elegant and useful educational tool, however.
A few minor issues:
There something odd about the sentence on lines 84-86. Maybe a “that” instead of the colon after “are” would read better.
Line 106: delete the second “bacteria”
Line 113: the symbols in this formula are not defined. In Table 1, “mu” is used to denote mutation rate.
Some of the layout on pages 7 and 9 is messed up.
Author Response
This paper describes a program “Cocktail” for simulating bacteriophage-host interactions in a controlled environment. The program is easy to use, with a simple and attractive graphical interface, and would be useful for educational purposes and potentially to generate testable hypotheses about the phage-host dynamics in a real-world environment.
The paper is well written and clearly laid out. I am not qualified to assess the mathematical model itself, but it seems well described and I will assume that the mathematical foundation is sound.
The main flaw of the paper is that there is no attempt to compare the results of the simulation to an actual experiment. How does the program replicate phage-host dynamics in an real chemostat? Even if the author does not have access to a chemostat in his lab or institution, there must be examples in the literature to which the program’s output can be compared. As the author himself points out: "Hypotheses will always need to be tested experimentally" (line 82). The paper would be strengthened considerably if the author did just that.
Compliance with earlier work can be checked by running the data files provided as supplementary material; they all relate to previously published work. The new data file Bohannan_Lenski_1997_Fig 3B.ctl contains settings for simulation of the first 200 hours of a chemostat experiment presented in the Bohannan & Lenski 1997 figure 3B but the resource density is set to 1.0 instead of 0.5 mg/L (ug/ml) in the chemostat. The data used is also referred to in table 1 in Abedon 2009 as well as used with modifications to produce figure 2F in Abedon 2009. The output of a Cocktail run of this file has been added as figure 2A in the manuscript. Here, the time step size is set to 1 minute instead of 3 in Bohannan & Lenski 1997.
A common misunderstanding of mathematical modelling is that is should perfectly match reality. A model that does this becomes enormously large and the results cannot be understood in exactly the same way as reality cannot be understood. Models are made for a completely different reason. The aim is to study the impact of a single variable, or a limited number of variables, against a backdrop of a fixed set of parameters. This makes it possible to infer hypotheses which can be experimentally tested. Models are never made to corroborate reality, e.g. the results of chemostat experiments, as implied by reviewers of this manuscript. The resulting parameter values from a model relies on many variables, but you can get the same output values from different combinations of input variables. E.g. if you observe bacterial phage resistance in a chemostat, how would you know if phage resistant cells are present from the beginning, formed de novo or released from biofilm without doing real in vivo experiments? A mathematical model would not provide the answer.
Running a modelling program with different input values until you get to what was observed in a chemostat experiment is not science. Running a modelling program to get testable predictions and hypotheses is. This is why the statement on line 82 reads “Hypotheses will always need to be tested experimentally” and not “Experiments will always need to be simulated theoretically”.
The results section is vague as to the source of the numbers input to the program: “The example describes two phages in a cocktail with different latent periods.” What two phages? Where do the default values come from? I would like to see a table of phages and hosts with the actual values for absorption rates, latent periods, burst sizes, growth rates, mutation rates etc. and how the program performs with these known parameters. Otherwise, there is no way to know whether the output from the program is just a pretty plot, or actually reflecting phage-host dynamics in a chemostat, let alone in a complex, real-world scenario.
More examples have been added demonstrating the capabilities of the program. Parameter settings are what is typically observed for phages and bacteria in all cases. Data on parameters can be found in the Supplementary materials files. A figure legend has been added to further explain the outputs. Figure 2A shows the Cocktail output of the data from the chemostat experiment from Bohannan & Lenski 1997 described above.
Although the manuscript does state explicitly that there are limitations to the program, it could state more clearly at the outset that the program models phage-host interactions in a chemostat. (For example, add “…in a chemostat” at the end of the first sentence of the abstract.) It is probably a stretch to imagine that the modeling will be relevant to an experimental phage therapy setting, and I would tone down this angle; I do think that the program could serve as an elegant and useful educational tool, however.
The suggested ending of the first sentence of the Abstract has been added and a new sentence added to the Introduction section (from line 75). “Mathematical modelling of phage infection in a chemostat may however set boundaries to what can be expected while it can reflect the dynamics under ideal conditions (e.g. constant nutrient supply and agitation). The results of mathematical modelling can never replace experiments. This is pointed out (line 53 -). Modelling merely results in a better understanding (line 61 - ). The program does not consider in vivo dynamics (line 74 -).
Referring to phage therapy in the introduction can be justified in that if theoretical characters of a phage are known, it may be discarded or other parameters changed if it is planned to be used in therapy. For example, a phage like PhiX174 would never be considered while its adsorption rate is E-14. Such a low adsorption cannot reduce bacterial titres if these are low (Nilsson AS. Pharmacological limitations of phage therapy. Ups J Med Sci. 2019 Nov;124(4):218-227. doi: 10.1080/03009734.2019.1688433. Epub 2019 Nov 14. PMID: 31724901; PMCID: PMC6968538.). This can be theoretically understood by running the Cocktail program.
A few minor issues:
There something odd about the sentence on lines 84-86. Maybe a “that” instead of the colon after “are” would read better.
The new reading of the sentence (78 -) is “The basic condition is a bacterial population, growing in a vessel in a constant volume of nutrient, which can become infected by phages at varying titres and times”.
Line 106: delete the second “bacteria”
Deleted
Line 113: the symbols in this formula are not defined. In Table 1, “mu” is used to denote mutation rate.
Some of the layout on pages 7 and 9 is messed up.
These are the original symbols used by Monod. The sentence “where the growth rate, µ, is a function of the maximum growth rate, µmax, a constant, K, and the concentration of nutrients, s” is added after the sentence. The change of symbols in the program is given in the following sentence (line 101).
The template cannot handle formats properly. The margins got changed after uploading the file. I have tried to adjust the formatting on pages 7 and 9 by adjusting the margin after page 1.
Reviewer 2 Report
In this article, the author presents an app-like software that aims at allowing users with no programming background run simulations of phage-bacteria interactions under a diversity of scenarios. I have never reviewed an article presenting software before, so I am not sure of whether to comment on the scientific merit of the article and its flow. It reads as a manual more than as an article, with a bit more background than a regular documentation file, which may be the intention of the author. I think, however, that is a missed opportunity to show the strengths of the framework and to "sell" the idea of the software to the academic audience of the journal. My comments below are intended to help make this a more academic article, as well as to improve the framework itself to reach a wider and more targeted audience.
1) The framework: I found the framework very interesting, and with a lot of potential. It does consider lots of different aspects of the interactions between bacteria and phage, but that is at the same time a strength and a weakness of the framework. The overall justification that the author presents is the use for hypothesis development in (clinical) phage therapy; however, the (many) simplifications needed for the model prevents it from being useful in such a complicated environment. My first impression before even starting reading is that at last someone is making available host-virus interaction models to students that do not know programming; but unfortunately the framework is too complicated as is to be educational.
My recommendation is to subdivide the framework into several interfaces that allow the user to focus on the aspects they are interested in. I understand that now the "mode" and other options to choose among kind of do that, but there are too many combinations and it is easy to get lost in the complexity. If there were either presets or, as I say, different interfaces, a user could for example choose "only ecology", for a simple scenario of 1 phage + 1 bacterium populations; "evolution of resistance", for a scenario in which there's either one or two phages and resistance can happen, etc. Right now, the user needs to understand and navigate several options to get those very educational and illustrative cases.
In a related point, there is a "default" set of parameters, but they do not inform the user of what real system they correspond to. I would actually add some combinations that consider the typical bacteria and typical phage used in experiments or phage therapy. For example, a user should be able to choose "E. coli" or "T-4 virus", and that would populate their respective parameters. The reason is that uneducated users may choose values that are reasonable for one species but unreasonable for others, and end up with a parametrization for host and virus that independently make sense but together does not, and therefore with results that are not representative of any realistic system.
I would be imperative as well that the user were made aware of the simplifications and assumptions that underlie the modeling options, so that they know whether they are applicable to whichever system they plan to use the framework for. For example, the interfaces and parametrization presets above could come with a warning that described the system, and what is left out.
One of those assumptions is the way superinfection is considered. Superinfection is negligible for many phages, which have mechanisms by which they can get rid of secondary infecting competing phage. However, if I understand well there is no way to "switch off" superinfection in the framework. If that were the case, phage could still attach to cells, but if the cell is already infected then the "superinfecting" phage does nothing. On the other hand, I am having a hard time with how "active" superinfection is dealt with. I understand and agree with the description for one cell, but I AB represents the population of (super)infected cells, each of which has been infected at a different time. Without keeping track of when each cell was infected, how can the framework know how much of the latent period has been consumed when superinfection happens (and therefore how can it be decided whether one or the other type of virus "wins" this cell)? Also, because there is a distribution of infection times, I cannot understand why IAB(t-lLOSING_STRAIN) becomes zero (since, again, for some cells within IAB it will be virus A lysing the cell and for others, B). Has the author considered this?
One assumption that I did not understand is that infected cells mutate (third term of eqs 6&7). Why can they mutate if they cannot reproduce?
And one assumption/aspect that may not be too important in realistic scenarios and that complicates considerably equations and explanations is multiple phage attachment to once cell. I think that part is one of the ones that could be safely removed for all users but, if it is to stay in the framework, the explanation (and way to refer to it in the software) needs to be considerably improved. In the manuscript, for example, it is really difficult to understand what the section is about almost until half way in the section.
Other important thoughts:
- If stochasticity is "switched on", several replicates are needed (as the author indicates). Will the user be able to see then the replicate-specific dynamics, the average over replicates, or both? The average is important, but given that oscillations may occur, it needs to be shown with a warning.
- PA and PB have the wrong units. According to eq11, they are the number of new adsorbed viruses per unit time, since delta is a rate (and that's why they are added directly to eqs 13 and 14). Please correct table 1 accordingly.
- The time step should not be an option to choose for the user, given that the choice has consequences for the accuracy of the framework.
2) Improving the article:
My first recommendation is to widen the scope of the framework. By focusing the introduction so much on (clinical) phage therapy, the author alienates lots of potential users for the framework, and skeptical readers will immediately point out that the framework is way too simplistic to be of any direct use for phage therapy. I would just add more examples, and definitely add explicit mentions as to how the framework can be used for educational purposes. Just as a reminder, though: phage therapy also refers to the use of phages to kill bacteria affecting plants.
I would indeed maybe add a structured section aimed at educators or other potential users, with one example of use per subsection. That would help "sell" the idea, and would definitely make the manuscript less of a documentation file.
In general, I would recommend structuring the sections in a more organized way, at least one that allows for the presentation of the different "modes" in a more systematic way. Right now, the modes are presented as they are needed, but since they are so many, it gets very messy and confusing very quickly. I would recommend also a dedicated table just to explain what the different modes are.
In a related point, I would definitely define what "refuge cells" means. The term is used many times before it is defined, which is very confusing.
Also, some parts are misleading or not correct. for example, L84 seems to talk about a chemostat as a possible environment but it is the chosen environment for the framework (i.e. it's non-optional); L110 talks about logistic growth of bacteria in a chemostat, but that would be the case if there is no viruses, and Monod by itself does not produce logistic growth (only if competition or an explicit resource are accounted for). There are several instances in which the author jumps from continuous time (ordinary differential equations, which define the model) to discrete time equations (L134-136, eqs15&16) to justify the shape of a specific term, which is confusing because the reader does not know whether those equations are part of the model. And N should be defined as the addition of all cells, which is not said explicitly.
Other comments:
- viruses are phenotypically variable as well (for the same strain, the same trait can take different values depending on the host, which is termed "viral plasticity" by some authors; the author says that only hosts are, but then points to the possibility of a wide latent period distribution, actually).
- when describing equations, I would make sure to refer to explicit terms (first term, second term, etc) when presenting the process it represents.
- L192: swap "can" and "of course" to form cannot.
- "Time interval" is misleading because it would normally be interpreted as the duration of the interaction. Call it time step size like in the framework.
In summary, the framework is a great and timely effort that has lots of potential, but the manuscript should be written in a less "documentation" and more contextual way, and the framework itself needs some tweaking to make it more user-friendly so that it can reach the (wide) audience it is intended for.
Author Response
In this article, the author presents an app-like software that aims at allowing users with no programming background run simulations of phage-bacteria interactions under a diversity of scenarios. I have never reviewed an article presenting software before, so I am not sure of whether to comment on the scientific merit of the article and its flow. It reads as a manual more than as an article, with a bit more background than a regular documentation file, which may be the intention of the author. I think, however, that is a missed opportunity to show the strengths of the framework and to "sell" the idea of the software to the academic audience of the journal. My comments below are intended to help make this a more academic article, as well as to improve the framework itself to reach a wider and more targeted audience.
Mathematical modelling of phage – bacteria interactions is an established line of research and not invented by me. The Cocktail program collects many of the recent ideas into an easy to use program for informed users.
1) The framework: I found the framework very interesting, and with a lot of potential. It does consider lots of different aspects of the interactions between bacteria and phage, but that is at the same time a strength and a weakness of the framework. The overall justification that the author presents is the use for hypothesis development in (clinical) phage therapy; however, the (many) simplifications needed for the model prevents it from being useful in such a complicated environment. My first impression before even starting reading is that at last someone is making available host-virus interaction models to students that do not know programming; but unfortunately the framework is too complicated as is to be educational.
I have not justified any hypothesis development aimed at (clinical) phage therapy. Please point out where you find this “justification”. In the introduction, the first paragraph is about the risen interest for phage therapy as a motivation for a more intense study of phage infection kinetics.
”The results of mathematical modelling can never replace experiments. This is pointed out (line 53 -).
“hypotheses about the population dynamics of bacteria and phages during a phage infection and not exact predictions” (line 73).
The program does not consider in vivo dynamics (line 74 -).
“Hypotheses will always need to be tested experimentally” (line 75)
“it is also possible to analyse more complex problems and formulate hypotheses that later can be evaluated experimentally” (line 277)
“The main purpose of the program is to generate hypotheses on bacteriophage infection dynamics that can be experimentally tested” (line 355)
What I have stated is on the contrary:
Modelling merely results in “a better understanding of the complex dynamics during phage therapy” (line 61 - ).
A common misunderstanding of mathematical modelling is that is should perfectly match reality. A model that does this becomes enormously large and the results cannot be understood in exactly the same way as reality cannot be understood. Models are made for a completely different reason. The aim is to study the impact of a single variable, or a limited number of variables, against a backdrop of a fixed set of parameters. This makes it possible to infer hypotheses which can be experimentally tested. Models are never made to corroborate reality, e.g. the results of chemostat experiments, as implied by reviewers of this manuscript. The resulting parameter values from a model relies on many variables, but you can get the same output values from different combinations of input variables. E.g. if you observe bacterial phage resistance in a chemostat, how would you know if phage resistant cells are present from the beginning, formed de novo or released from biofilm without doing real in vivo experiments? A mathematical model would not provide the answer.
Running a modelling program with different input values until you get to what was observed in a chemostat experiment is not science. Running a modelling program to get testable predictions and hypotheses is.
My recommendation is to subdivide the framework into several interfaces that allow the user to focus on the aspects they are interested in. I understand that now the "mode" and other options to choose among kind of do that, but there are too many combinations and it is easy to get lost in the complexity. If there were either presets or, as I say, different interfaces, a user could for example choose "only ecology", for a simple scenario of 1 phage + 1 bacterium populations; "evolution of resistance", for a scenario in which there's either one or two phages and resistance can happen, etc. Right now, the user needs to understand and navigate several options to get those very educational and illustrative cases.
Mathematical modelling of simple cases of phage infection has been done for decades. If needed, I can expand the reference list with several dozens of other articles relating to one phage - one host modelling, but a few is mentioned in the manuscript. The purpose with the program is to offer a simple way of analysing more complicated cases, and the intended use is by scholars. This is the reason behind publishing in a scientific journal.
In a related point, there is a "default" set of parameters, but they do not inform the user of what real system they correspond to. I would actually add some combinations that consider the typical bacteria and typical phage used in experiments or phage therapy. For example, a user should be able to choose "E. coli" or "T-4 virus", and that would populate their respective parameters. The reason is that uneducated users may choose values that are reasonable for one species but unreasonable for others, and end up with a parametrization for host and virus that independently make sense but together does not, and therefore with results that are not representative of any realistic system.
The default values are mainly from Lenski 1988 as clearly stated in the manuscript. These parameter values are average values commonly found for both phages and bacteria. Entering a particular parameter value outside of what is known for phages or bacteria will evoke an error. Hovering over an input box will display the allowed range.
All E. coli, and all T4 phages, are different and it is therefore meaningless to supply sets of data for “E. coli” or “T4” modelling. Parameter values are usually known by the scholars that will use and benefit from the program. The program is not intended for uneducated users.
I would be imperative as well that the user were made aware of the simplifications and assumptions that underlie the modeling options, so that they know whether they are applicable to whichever system they plan to use the framework for. For example, the interfaces and parametrization presets above could come with a warning that described the system, and what is left out.
A model is always built on simplifications, this program is no exception. Some simplifications are stated in the manuscript to remind the user about this. Well-known simplifications are however not discussed while the researchers who can conceivably use the program should know about these. See also “A common misunderstanding…” above.
One of those assumptions is the way superinfection is considered. Superinfection is negligible for many phages, which have mechanisms by which they can get rid of secondary infecting competing phage. However, if I understand well there is no way to "switch off" superinfection in the framework. If that were the case, phage could still attach to cells, but if the cell is already infected then the "superinfecting" phage does nothing. On the other hand, I am having a hard time with how "active" superinfection is dealt with. I understand and agree with the description for one cell, but I AB represents the population of (super)infected cells, each of which has been infected at a different time. Without keeping track of when each cell was infected, how can the framework know how much of the latent period has been consumed when superinfection happens (and therefore how can it be decided whether one or the other type of virus "wins" this cell)? Also, because there is a distribution of infection times, I cannot understand why IAB(t-lLOSING_STRAIN) becomes zero (since, again, for some cells within IAB it will be virus A lysing the cell and for others, B). Has the author considered this?
Superinfection exclusion is more common among temperate phages and not explicitly considered in the program. The model treats superinfection as a competition between phages both being able to infect the bacterium (the model would not make sense otherwise). As stated in the manuscript: “This results in that becomes zero for a bacterium when becomes positive. One of and subsequently always becomes zero. A bacterium infected with phage B at time tB and superinfected with phage A at time tA will lyse and produce phages of type A only if tA + lA is shorter than the remaining time to lysis caused by phage B, i.e. tA + lA < tA + lA – tB + lB = tA < tA – tB + lB.” This takes up a lot of computer memory as the program needs to keep track of when a bacterium got infected and by which phage as well as when the same bacterium got infected by a second different phage and the time left to lysis by the first phage compared to the time left to lysis by the second phage. As stated in the manuscript, this leads to (part of) the interference between phages. The sentence “Other interference between phages, e.g. by actively impeding the other phage’s transcription or replication, is not considered.” is added to the paragraph. Such interference probably exist for virulent phages, but this has not been shown. Nevertheless, if a user wants to simulate superinfection exclusion it is just to set the burst size of the superinfecting phage to zero. This will result in loss of the phage due to adsorption but no replication of the phage in host cells.
One assumption that I did not understand is that infected cells mutate (third term of eqs 6&7). Why can they mutate if they cannot reproduce?
Thanks for noting this. This is a cut-and-paste error and the terms have been deleted.
And one assumption/aspect that may not be too important in realistic scenarios and that complicates considerably equations and explanations is multiple phage attachment to once cell. I think that part is one of the ones that could be safely removed for all users but, if it is to stay in the framework, the explanation (and way to refer to it in the software) needs to be considerably improved. In the manuscript, for example, it is really difficult to understand what the section is about almost until half way in the section.
Multiple phage attachment to a cell is important in realistic scenarios if phage titres are high and the time step size long (see also answer to why the time step size can be set below). Only a basic knowledge about probability distributions is needed for understanding of the rationale behind adding another phage adsorption model. This addition has been discussed before and should not be new to the reader. Reference to Abedon 2009 is added to line 193.
Other important thoughts:
- If stochasticity is "switched on", several replicates are needed (as the author indicates). Will the user be able to see then the replicate-specific dynamics, the average over replicates, or both? The average is important, but given that oscillations may occur, it needs to be shown with a warning.
The average is of course given by running Cocktail with the deterministic setting. The stochastic setting, as well as the option to round off cell titre values below zero, is to facilitate simulating real conditions with real values. Mutations happen randomly as cells divide but a fraction of a cell cannot divide and there cannot be a fraction of a mutated cell. This only has to be considered if cell numbers are small.
- PA and PB have the wrong units. According to eq11, they are the number of new adsorbed viruses per unit time, since delta is a rate (and that's why they are added directly to eqs 13 and 14). Please correct table 1 accordingly.
Table 1 has been updated. Phage titres should be denoted A and B.
- The time step should not be an option to choose for the user, given that the choice has consequences for the accuracy of the framework.
Other published models have had different and fixed time steps. The option to be able to choose different time steps makes it possible to compare the results from Cocktail to previously published results. There is also a time – accuracy trade-off background, explained in the manuscript, to the possibility to set different time steps.
2) Improving the article:
My first recommendation is to widen the scope of the framework. By focusing the introduction so much on (clinical) phage therapy, the author alienates lots of potential users for the framework, and skeptical readers will immediately point out that the framework is way too simplistic to be of any direct use for phage therapy. I would just add more examples, and definitely add explicit mentions as to how the framework can be used for educational purposes. Just as a reminder, though: phage therapy also refers to the use of phages to kill bacteria affecting plants.
The first paragraph of the introduction is the major reason why researchers are interested in phage infection kinetics. By reaching out to these, I hope to convey the complexity of phage therapy as this can be clearly shown by the Cocktail program. Many researchers, especially non-phage biologists (or even non-biologists) interested in phage therapy, have a very simplistic view of the subject matter. See also previous comments. Some example files are indeed added as supplementary material, but the problem with just about any example file is that it would represent only a tiny fraction of the parameter space and could lead users to draw wrong conclusions. There is unfortunately no simple standard cases that would fit as examples. More examples have been added though, as shown in figures 2A – D.
I would indeed maybe add a structured section aimed at educators or other potential users, with one example of use per subsection. That would help "sell" the idea, and would definitely make the manuscript less of a documentation file.
I am convinced that the intended users understand how to use the program. The program is not intended for the general public or undergraduate students as they would have no use of it. If teachers and students want to delve into phage infection kinetics they would be better off studying it by other even simpler software e.g. ODE solving programs like Mathematica or Matlab.
In general, I would recommend structuring the sections in a more organized way, at least one that allows for the presentation of the different "modes" in a more systematic way. Right now, the modes are presented as they are needed, but since they are so many, it gets very messy and confusing very quickly. I would recommend also a dedicated table just to explain what the different modes are.
The program model labels have been changed to better reflect how the model works. “Phage infection” was a remnant from the simpler version Cocktail 1.0 where secondary adsorption was not modelled. The label has now changed to “Primary adsorption” and “Secondary” has been added to the other label.
The structure of the presentation of the models and model settings now reads:
2.4 Model settings
New paragraph inserted
2.4.1 Primary adsorption: Standard model
2.4.2 Primary adsorption: Poisson
2.4.3 Resistance mutation
2.4.4 Refuge cells
2.4.5 Time step size
These paragraphs deal, in the same order as in the GUI, with the mathematical background and equations. It is also explained why a particular model is included. The intended user will notice that the structure, the mathematical background and the equations are treated as in other publications. The additions in Cocktail are discussed in the articles cited, but are not implemented in full in any other computer program. A table (table 2) has been added after a short introductory paragraph to Model settings 2.4.
In a related point, I would definitely define what "refuge cells" means. The term is used many times before it is defined, which is very confusing.
Mathematical modelling of phage – bacteria infection dynamics in a chemostat has been done for a very long time. The words “Refuge cells” is an established designation for cells that for any reason are refractory to phage infection. Any intended user of the program will know what “Refuge cells” means. It was coined (to my knowledge) by Levin B et al. in the 1977 paper I refer to and has been used ever since. References to paragraphs describing mutations and refuge populations have been inserted in line 142, after “Bacteria can…hide in a refuge”. The reader is also reminded later on “Bacteria can also move into a refuge population, Sr, at a rate of σ where they may or may not be adsorbed by phages” (line 156) ; “Cells in the refuge…is simulating the presence of either metabolically inactive cells or cells in biofilm and inhibited phage propagation” (line 164). The first line of section 2.4.4 reads “The refuge population is simulating the existence of metabolically inactive cells, with the “Planktonic” setting, or cells forming biofilm with the last-in-first-out (“LIFO”) setting.”.
Also, some parts are misleading or not correct. for example, L84 seems to talk about a chemostat as a possible environment but it is the chosen environment for the framework (i.e. it's non-optional);
Yes, it seems to be a chemostat that is described in line 78+. The sentence that follows reads: “Although the volume being constant in such a chemostat…”, and establishes that this is the case. Can there be any doubt about what the program is all about?
L110 talks about logistic growth of bacteria in a chemostat, but that would be the case if there is no viruses, and Monod by itself does not produce logistic growth (only if competition or an explicit resource are accounted for).
It is possible to run the program without infecting phages and with limited resources. This results in a logistic growth of the bacteria. Hence “In a closed system, while nutrients are consumed by the bacteria, their concentration decreases and growth slows. Seen over time, in such cases the bacterial growth becomes a logistic function of the concentration of nutrients”. My italics. After this sentence, I continue with the growth function of Monod. I never claim that you get a logistic growth in other cases.
There are several instances in which the author jumps from continuous time (ordinary differential equations, which define the model) to discrete time equations (L134-136, eqs15&16) to justify the shape of a specific term, which is confusing because the reader does not know whether those equations are part of the model. And N should be defined as the addition of all cells, which is not said explicitly.
Line 134 – 136 (now line 118 -). Note the use of an indefinite article in the preceding sentence. The wording refers to any case of titre change when bacteria decay from natural reasons. Immediately after the formula is given, it is explained what N stands for. It is also obvious to the reader that these are basic equations that serve as foundations (arriving at the gamma parameter inserted into the equations for all bacteria) to the following while they are embedded in the text without equation numbers.
Equations 15 and 16 are explained in the text just for clarity. They are indeed part of the model as they are added into equation 17. Equation numbers have been added to the text.
Other comments:
- viruses are phenotypically variable as well (for the same strain, the same trait can take different values depending on the host, which is termed "viral plasticity" by some authors; the author says that only hosts are, but then points to the possibility of a wide latent period distribution, actually).
The varying latent period referred to in the text depends entirely on the nutritional condition of individual hosts. Bacteria are single cells and cannot be compared to eukaryotes. Transcription and replication by some cells is faster than in other cells, but it is not a property of the phages. Some virulent phages however harbour genes for an extended latency, resulting in dormancy or stalled transcription of pertinent genes, as this may be adaptive under certain conditions (e.g. the original T4 has a system for this). Neither of these are considered in the program models. The latent period is the average from many infections.
- when describing equations, I would make sure to refer to explicit terms (first term, second term, etc) when presenting the process it represents.
Parameters in equations are given in table 1 and the terms are explained in order after each equation. If “first term, second term”, and so forth, is added, it would only result in annoying repetitions and not contribute to the readability.
- L192: swap "can" and "of course" to form cannot.
The use of the somewhat colloquial “of course not” is changed to an equivalent more literary expression.
- "Time interval" is misleading because it would normally be interpreted as the duration of the interaction. Call it time step size like in the framework.
The sub-clause “set either as one minute or as 30, 15 or a five seconds time step size” has been added to the first sentence of 2.4.5. “Time interval” is the established expression in ODE solving theory.
In summary, the framework is a great and timely effort that has lots of potential, but the manuscript should be written in a less "documentation" and more contextual way, and the framework itself needs some tweaking to make it more user-friendly so that it can reach the (wide) audience it is intended for.
The submitted manuscript is the documentation for the program. It refers to other work on modelling and introduces some new concepts. The intention is not to present a funny tool for a wide audience who do not understand the complexity, but phage biologists interested in getting some theoretical background to experimental planning e.g. for setting limits to phage therapy.
Reviewer 3 Report
The purpose of the Cocktail program is to model the infection dynamics of one, or a combination of two, phage(s) infecting a bacterial species under varying relevant parameter settings, in the latter case also to some extent the interference between two phages infecting at the same time. The aim is to supply an easier way to carry out modelling in phage infection biology, for a better understanding of the complex dynamics during phage therapy. This work is meaning and potential in this field.
1 Authors should show an outline of this work.
2 The language should be further improved.
3 There are too much formulations. Authors should show the relationships among them.
4 The some important references should be discussed in this work.
Author Response
The purpose of the Cocktail program is to model the infection dynamics of one, or a combination of two, phage(s) infecting a bacterial species under varying relevant parameter settings, in the latter case also to some extent the interference between two phages infecting at the same time. The aim is to supply an easier way to carry out modelling in phage infection biology, for a better understanding of the complex dynamics during phage therapy. This work is meaning and potential in this field.
1 Authors should show an outline of this work.
What kind of outline?
2 The language should be further improved.
Where?
3 There are too much formulations. Authors should show the relationships among them.
The formulations are unfortunately in essence what the manuscript is all about. The relationship is given by the ODEs being solved.
4 The some important references should be discussed in this work.
More specifically, what needs to be discussed?
Round 2
Reviewer 1 Report
This work can be accepted.
Reviewer 3 Report
This work can be accepted.